# Exploring the Impact of Climatic Variables on Arecanut Fruit Rot Epidemic by Understanding the Disease Dynamics in Relation to Space and Time

**DOI:** 10.3390/jof8070745

**Published:** 2022-07-19

**Authors:** Balanagouda Patil, Vinayaka Hegde, Shankarappa Sridhara, Hanumappa Narayanaswamy, Manjunatha K. Naik, Kiran Kumar R. Patil, Hosahatti Rajashekara, Ajay Kumar Mishra

**Affiliations:** 1Department of Plant Pathology, University of Agricultural and Horticultural Sciences, Shivamogga 577255, Karnataka, India; swamyareca@gmail.com (H.N.); manjunaik2000@yahoo.co.in (M.K.N.); 2Division of Crop Protection, ICAR-Central Plantation Crops Research Institute, Kasaragod 671124, Kerala, India; vinayaka.hegde@icar.gov.in; 3Center for Climate Resilient Agriculture, University of Agricultural and Horticultural Sciences, Shivamogga 577255, Karnataka, India; sridharas1968@gmail.com; 4Department of Agricultural Economics, University of Agricultural and Horticultural Sciences, Shivamogga 577255, Karnataka, India; kiranecon@gmail.com; 5Division of Crop Protection, ICAR-Directorate of Cashew Research, Puttur 574202, Karnataka, India; rajashekara.h@icar.gov.in; 6Khalifa Centre for Genetic Engineering and Biotechnology, United Arab Emirates University, Al Ain P.O. Box 15551, United Arab Emirates

**Keywords:** arecanut, fruit rot disease, spatio-temporal pattern, ordinary kriging, climatic indicators, epidemiology, spatial dependency, GLM

## Abstract

To understand the spatio-temporal dynamics and the effect of climate on fruit rot occurrence in arecanut plantations, we evaluated the intensity of fruit rot in three major growing regions of Karnataka, India for two consecutive years (2018 and 2019). A total of 27 sampling sites from the selected regions were monitored and the percentage disease intensity (PDI) was assessed on 50 randomly selected palms. Spatial interpolation technique, ordinary kriging (OK) was employed to predict the disease occurrence at unsampled locations. OK resulted in aggregated spatial maps, where the disease intensity was substantial (40.25–72.45%) at sampling sites of the Malnad and coastal regions. Further, Moran’s I spatial autocorrelation test confirmed the presence of significant spatial clusters (*p* ≤ 0.01) across the regions studied. Temporal analysis indicated the initiation of disease on different weeks dependent on the sampling sites and evaluated years with significant variation in PDI, which ranged from 9.25% to 72.45%. The occurrence of disease over time revealed that the epidemic was initiated early in the season (July) at the Malnad and coastal regions in contrary to the Maidan region where the occurrence was delayed up to the end of the season (September). Correlations between environmental variables and PDI revealed that, the estimated temperature (T), relative humidity (RH) and total rainfall (TRF) significantly positively associated (*p* = 0.01) with disease occurrence. Regression model analysis revealed that the association between T_max_, RH1 and TRF with PDI statistically significant and the coefficients for the predictors T_max_, RH1 and TRF are 1.731, 1.330 and 0.541, respectively. The information generated in the present study will provide a scientific decision support system, to generate forecasting models and a better surveillance system to develop adequate strategies to curtail the fruit rot of arecanut.

## 1. Introduction

Arecanut (*Areca catechu* L.) is a tropical industrial tree species that plays a crucial role in providing livelihood and economic security to millions of people in India. It is one of the most predominant industrial crops in Southern and South-East countries, including India, China, Indonesia and Malaysia [1]. Arecanut cultivation generates direct and indirect job opportunities, creates a diverse array of materials to several small- and medium-scale industries and plays a critical role in conservation of the soil and ecosystem [2]. Even though the arecanut cultivation is location-specific, its consumption is widespread throughout the country. The arecanut-based industries primarily act as a major sole source of income to six million arecanut farmers and stakeholders [3]. Evidence has been suggested that arecanut has different pharmacological benefits, such as anti-parasitic, bacterial, fungal, inflammatory and analgesic effects [4]. 

Arecanut is majorly grown in the southern and northeastern parts of the country, which includes Karnataka, Kerala, Tamil Nadu, West Bengal and Assam, which shares 90% of the total area and production [5]. Although the arecanut production is restricted to a few states, the major industrial products are globally shared, and about 700 million consume arecanuts world-wide [6,7]. India contributes about 57% of the global arecanut production and the remaining 43% was shared among China, Bangladesh, Myanmar and other countries. Irrespective of the arecanut growing regions, nuts were generally harvested upon maturity in the December–January months. Despite arecanut having industrial importance, the maximum potential is highly prone to attack by a wide range of biotic and abiotic stresses at all the growth stages. Arecanut production is majorly impaired by various fungal diseases. Among them, fruit rot stands at the topmost position, which results in the complete death of the palms either individually or for the whole plantation, which accounted for about 10–90% of the potential production [8]. Losses were over 75% or complete damage of the palm in Karnataka during 1910 [9] and further resulted in a significant economic loss of 15–20% in 1918 at Kerala [10,11,12], 50–90% in 1956 at parts of Malabar region [13] and 72–350 kg nuts ha^–1^ in 1985 in North Canara [14] was estimated.

The occurrence of fruit rot disease (FRD) is primarily confined to traditional and regularly growing tracts belongs to Karnataka, Kerala, Tamil Nadu, West Bengal, Assam, parts of Andaman Islands and Meghalaya states. Recently, arecanut cultivation is continuously increasing due to its higher industrial and economic benefits in non-traditional growing tracts (Maidan regions), resulting in the spread and survival of the pathogen in these areas. For the first time, fruit rot was reported from the erstwhile Mysore province [15] and later from the endemic areas of Dakshina Kannada (South Canara) and Uttara Kannada (North Canara) in Karnataka state and disease-prone coastal tracts (Malabar and Cochin) of Kerala [9]. Recently, the disease has been increasing over a period of time due to the buildup of inoculum load within the gardens and micro-climatic conditions favoring disease spread and development [9,16]. 

The occurrence of a pathogen, *Phytophthora meadii*, which McRae (1918) associated with disease, was documented from arecanut-growing areas of Kerala and Karnataka. *P. meadii* is an oomycetous hemi-biotrophic fungus that has harmful effects on arecanut by producing various symptoms at different stages of the palms [17,18,19]. The knowledge about the spatio-temporal patterns and the spread of the disease in disease-prone regions will provide valuable information to design suitable approaches for effective management of the disease [20]. Studies conducted to understand the spatio-temporal patterns of arecanut FRD and factors responsible for its spread are lacking. Information on the epidemiology and impact of weather parameters on FRD in these regions play a critical role in managing this disease due to the development of new virulent pathotypes of hemi-biotroph oomycete fungus *P. meadii* in the recent past [21,22].

Several factors might have contributed to the increased buildup of the disease, such as establishing new arecanut gardens in new regions, cultivating susceptible varieties across these areas and the unexpected change in climate and the frequent devastating cyclones [23]. Apart from these, the infection ability of *P. meadii* on arecanut strongly depends on the virulence of pathogen that prevailed across the regions of arecanut cultivation.

Earlier studies on epidemiology and the survival ability of the pathogen revealed that *P. meadii* causing fruit rot perpetuates as a dormant mycelium either in the treetop on infected and dried bunches or rotted buds present on affected palms or in the top soil in the form of oospores [9,11,14,19,24]. However, precise information on the epidemiological cycle, development of epidemics and their mode of spread needs to be assessed through appropriate experimentation. The initial inoculum and intra-plot microclimate play a pivotal role in the development of epidemics. 

Hence, there is a need to understand the level of the primary inoculum and its distribution between different survival sites. The description of the fruit rot epidemics regarding space and time scales in arecanut-growing areas may provide valuable information on the pathogen dissemination and epidemiological consequences. These kinds of studies are lacking at a regional scale in major arecanut-growing areas. Similar studies have been conducted in other plantation crops to describe the spatio-temporal patterns of black pod rot in cocoa at the regional scale in Cameroon. Recently, the spatial and temporal patterns of black pod rot disease were investigated in cocoa farms where the pathogen was already well established [25,26]. Further, experiments have mainly concentrated on point of infection on the pods by the pathogen in Nigeria [27], Brazil [28], the Solomon Islands [29] and the Ivory Coast [30].

The present study aimed to generate information on the spatio-temporal dynamics of FRD over different arecanut growing regions of Karnataka, India and to identify the environmental factors favoring the disease initiation and development. This study also aimed at assessing the possible impacts of climatic indicators on the regional occurrence of fruit rot epidemics as an aid for the adoption of efficient management tactics.

## 2. Materials and Methods

### 2.1. Ethical Declaration

No specific permission was required for the proposed nature of work. At every monitored sampling site, the arecanut growers permitted us to collect FRD-infected samples and to perform the sampling process. The investigation conducted during two consecutive years (2018 and 2019) did not involve any endangered or protected species of plants or animals. The fixed-plot surveys were conducted across the varied arecanut growing regions of Karnataka, and subsequent investigations were conducted at the Indian Council of Agricultural Research-Central Plantation Crops Research Institute, Kasaragod, Kerala, India.

### 2.2. Description of the Study Area

The field sampling and disease scoring was performed in three varied arecanut growing regions of Karnataka, India viz., coastal, Malnad and Maidan regions, respectively (Figure 1). The patterns of FRD regarding space and time scales were examined through continuous monitoring for disease (fixed-plot) in two successive seasons (Kharif 2018 and 2019). The description and particulars of the experimental area is provided in Table 1. The sampling points were selected randomly by considering uniformity in the PDI over a large area, and their distribution was chosen based on disease prevalence. 

The agro-climatic regions were chosen based on their variations in climate and topographic profiles. Nine sampling sites were selected from each region to assess the disease intensity. The arecanut palms were 20–45 years old and were planted at a spacing of 2.7 × 2.7 m (1371 palms hectare^−1^) in all the plantations considered for this study. The gardens were managed using no-tillage practices, and weed control was achieved with herbicidal treatments or manually.

### 2.3. Estimation of Disease Variables

The disease intensity of FRD in each garden was assessed weekly by visual observation of typical symptoms on palms along the rows throughout the entire garden. The locations of each diseased palm were mapped on the average monthly determined observations. Estimation of disease was done on randomly chosen and tagged fallen nuts per palm (*n* = 50) across the 18 weeks and then converted into the PDI. The PDI of fruit rot of arecanut was determined by the standard Formula (1) and disease rating scale (1–6) developed by Sastry and Hegde [31] as shown in Table 2.
(1)$$Pe\mathrm{rcent}\;\mathrm{Disease}\;\mathrm{Intensity}\;\left(\mathrm{PDI}\right)=\frac{\mathrm{Sum}\;\mathrm{of}\;\mathrm{numerical}\;\mathrm{ratings}}{\mathrm{Total}\;\mathrm{number}\;\mathrm{of}\;\mathrm{palms}\;\mathrm{observed}\;\times \;\mathrm{Maximum}\;\mathrm{rating}}\times 100$$

The PDI observations were collected weekly from the June to September months of monsoon season, totaling 18 weeks of observations per site.

### 2.4. Conceptual Framework of the Study

A conceptual framework regarding the procedures used to describe the spatio-temporal analyses and processes utilized during the study is presented in Figure 2. It provides a succinct summary of the methodology applied in this investigation and consists of data retrieval, analysis and processing using GIST tools.

### 2.5. Temporal Progression and Statistical Analysis of FRD at the Regional Scale

The PDI data was analyzed using the Kruskal–Wallis test in ‘R statistical software (version R-3.6.1)’ to determine the statistical difference observed among the sampling sites (*n* = 9) in each region. Significant variation in PDI at sampling sites was assessed through box and whisker plots depicting the upper and lower quartiles with outliers as individual points.

Temporal analysis was performed to understand the progression of the disease over sampling points among each region based on observations of disease intensity at weekly intervals during two consecutive years. Column graphs were plotted by using PDI against standard weeks at sampling sites in each region then dataset was compared among the years of experimentation and plots were prepared using ‘Graph-pad Prism (Version 9.0.1, GraphPad Software Inc. San Diego, CA, USA)’.

#### Generalized Linear Model (GLM)

The dependent variable PDI is count data that failed to follow a normal distribution. The influence of predictors, such as arecanut growing regions (Maidan, Malnad and coastal), 18 weeks starting from the June month to the end of September month and time element (years) on PDI cannot be estimated through multiple linear regression. Hence, a generalized linear model (GLM) with a Poisson or negative binomial regression with log link was tried. 

Arecanut growing regions having three categories are captured through two dummies, i.e., D_1_ and D_2_. D_1_ takes the value ‘1’ and D_2_ ‘0’ for Malnad region, D_1_ = 0 and D_2_ = 1 for the coastal region, and the control group was represented by D_1_ = 0 and D_2_ = 0. Though sampling weeks considered was from the first week of June to the last week of September, the disease occurrence was apparent only from the sixth week. Hence, the 6th to the 18th weeks were considered for model analysis and captured through dummies. 

There were 13 categories; hence, 12 dummies were used, i.e., D_3_ = 1, D_4_ = 0, D_5_ = 0, D_6_ = 0, D_7_ = 0, D_8_ = 0, D_9_ = 0, D_10_ = 0, D_11_ = 0, D_12_ = 0, D_13_ = 0 and D_14_ = 0 to represent the sixth sample week. The same analogy was followed to create the dummy matrix for the sample weeks. The influence of the time element, i.e., 2018 and 2019, was captured through dummy variable D_15_ assuming a ‘0’ value for 2018 and ‘1’ for 2019. SAS software version 9.4M7 was used for the estimation of Poisson regression and negative binomial regression. 

The deviance expected to be 1 is called ‘equi-dispersion’; if it is more than one, called ‘over-dispersion and less than one, called ‘under-dispersion. In the present study, the condition of equi-dispersion was unmet with GLM Poisson regression with log link (deviance = 5.6). To combat this econometric problem of overdispersion, a generalized form of Poisson regression, i.e., negative binomial regression with log link was performed. The estimable form of GLM negative binomial regression with log link is given below.
Log (PDI %) = β_1_+ β_2_D_1_ + β_3_D_2_ + β_4_D_3_ + β_5_D_4_ + β_6_D_5_ + β_7_D_6_ + β_8_D_7_ + β_9_D_8_ + β_10_D_9_ + β_11_D_10_ + β_12_D_11_ + β_13_D_12_ + β_14_D_13_ + β_15_D_14_ + β_16_D_15_

PDI = Percentage Disease Intensity; D1 and D2 represent dummy variables for the Malnad and coastal regions, respectively; whereas, D3–D14 indicted the dummy variables from 6th to 18th weeks, and D15 described the dummy variable for year.

### 2.6. Spatial Dynamics of FRD at Regional Scale

The spatial patterns of FRD distribution in varied regions of Karnataka were characterized by employing geostatistical tools viz., ordinary kriging (OK) approach and spatial autocorrelation. First, the OK tool was used to create spatial maps to predict FRD patterns across the regions that helps to identify significant clusters of disease in arecanut-growing areas. Secondly, the identified disease-prone areas were optimized through autocorrelation analysis and these analyses were performed using ArcGIS 10.3 version software. The region-wise sample data collected were tabulated indicating the latitude, longitude and PDI. Since PDI data was *log* (*x* + 1) transformed to understand the distribution of data set.

The OK approach was employed to estimate the PDI values of the spatial phenomenon at non-sampled locations, such as the disease intensity of FRD at locations (X_1_, X_2_, …, X_n_) are (Z_1_, Z_2_,…, Z_n_). The objective of spatial data view is to predict the Z value at unsampled points of X. Theoretically, kriging approach was used to optimize the surface area to predict random parameter Z at different unsampled locations [32,33]. A surface map of disease intensity of fruit rot was generated with the help of robust OK technique and can be mathematically expressed as following Equation (2):(2)$$Z\left(x_{0}\right)=\sum_{i=1}^{n}\lambda_{i}z\left(x_{i}\right)$$
where z is the PDI at locations *x_i_* and *x*_0_; *n* is the number of neighbors taken into consideration; *x_i_* and *x*_0_ are the sites observations recorded; *λ_i_* are the weights; and *z* (*x_i_*) is the value of z at *x_i_* [34,35,36,37].

### 2.7. Spatial Autocorrelation

To assess the degree of spatial autocorrelation among the neighboring plots having infected palms, Moran’s *I* statistic was utilized to compute the spatial dependence between experimental sites in each region [38,39,40]. The neighborhood category was taken into consideration as that of the nearest neighbors among different locations [41,42,43]. The interpretation of the spatial autocorrelation output was performed through Moran’s *I* test and inferred using a *p*-value, which estimated the existence of significant spatial clusters. The following Equation (3) was used to determine the spatial dependency is as mentioned below:(3)$$I=Z\sum_{i=1}^{n}W-Z$$
where *I* is the statistic for site *i*, *Z* indicates the difference in PDI values at *I*, and *n* is the average PDI for three regions calculated separately. *W* is the weight matrix that, in this case, only considered neighbors that share a common border or vertex.

### 2.8. Monitoring of Climatic Variables

An attempt was made to investigate the effect of environmental variables on FRD epidemics on arecanut plantations. The PDI was recorded from a farmer’s garden and research stations in varied regions of Karnataka. Environmental variables, such as the minimum and maximum temperature (T_max_; T_min_, °C), first half (morning) and second half (evening) Relative Humidity (RHI; RHII, %), Total Rainfall (TRF, mm), Number of rainy days (RD) and Wind Speed (WS, km/h), of the sampling points were collected from the nearest meteorological observatory maintained by the Agricultural and Horticultural Research Station (AHRS) belonging to the University of Agricultural and Horticultural Sciences, Shivamogga, Karnataka, India as well as from the Karnataka State Natural Disaster Monitoring Centre (KSNDMC), Bengaluru, Karnataka, India for two consecutive years (2018 and 2019).

The daily data were converted into weekly averages for T_max_, T_min_, RHI, RHII and WS whereas weekly totals were considered for TRF and RD. The combined data from the sampling sites in each region during 2018 and 2019 were gathered to analyze the correlation between disease intensity and climatic factors using ‘R statistical software (version R-3.6.1)’. A correlation was performed to investigate whether the micro-climate conditions were similar within or between gardens.

Further, we conducted multiple linear regression model to estimate the relationship between FRD (dependent variable, PDI) and seven independent variables (T_max_, T_min_, RHI, RHII, WS, TRF and RD), which often called as predictors or explanatory variables. The dataset combined from 2018 and 2019 were considered to estimate the predictable impact of explanatory parameters on response variable and also used to infer causal relationship between both the variables. The regression analysis was conducted in ‘R statistical software (version R-3.6.1)’. Confidence and prediction intervals were displayed at 95%, which gives the range of likely values from the mean response and single observation.

## 3. Results

### 3.1. Disease Intensity of Fruit Rot on Arecanut in Karnataka

The pathogen infection was found throughout the sampling points (except few points in Maidan areas) over the regions with varying disease intensity (PDI), which significantly differed across the studied regions of Karnataka. A significant difference in PDI was observed among sampling points in the Malnad region, where S2 exhibited a higher disease proportion than other points. In contrast, sampling sites in the coastal region did not show considerable differences in disease intensity that indicated almost similar trend was observed across the region. Disparately, no disease was observed in few sampling plots of the Maidan region (S1, S7, S8 and S9) even though the remaining plots exhibited slight variation with minimum disease proportion (Figure 3).

### 3.2. Temporal Analysis of FRD

FRD progression over time in the Malnad region during both years substantially differed and was not consistent among the sampling plots for disease intensity (Figure 4). In all the plots, the disease intensity began during the sixth week of observation (second fortnight of July) in both seasons. Then, the disease progression linearly increased over weeks of observations and from the initiation of disease development. Further, it progressed consistently with the varied proportion of disease until the end of the season. 

Whereas, at the coastal region, the disease intensity was seen early (first week of July) compared to other regions irrespective of sampling sites where arecanut palms showed initiation of disease. Further, it was distributed virtually throughout the plots and neighboring places with different disease proportions (Figure 4). From the fifth week onwards, the disease was initiated tremendously up to the end of August. However, decreased disease intensity was noticed during the 15th to 18th week of disease monitoring.

Contrastingly, at the five experimental sites of the Maidan region, disease intensity progression seemed to exhibit a gradient, which was less evident at other regions (Figure 4). During both seasons, the disease to be initiated from the 12th week of observation and disease occurrence gradually increased over time with slight variation in disease proportion. In some of the Maidan region sampling sites, the disease was not noticed during both seasons after 18 weeks of observation.

### 3.3. Generalized Linear Model: Negative Binomial Regression with Log Link

We constructed a GLM model, which explained the FRD intensity through the proposed explanatory variables and then applied Akaike Information Criteria (AIC) to confirm the better fit of the GLM model: negative binomial regression with log link.

The estimates of the GLM negative regression model with log link and their standard errors are presented in Table 3 and Appendix A. The AIC was lower in negative binomial regression (5216) compared to Poisson regression (infinite). The other criteria that reiterated negative binomial regression for the PDIS are deviance factors. The residual deviance, similar to the residual sum of squares in usual multiple regression, is expected to be low/minimum. The residual deviance in the case of negative binomial regression was 1.4, which was substantially lower than Poisson regression with the deviance of 5.6. The results outlined below are pertinent to negative binomial regression. 

All the regression coefficients excepting dummy variables representing 14th, 15th, 16th and 17th sample weeks and time elements were statistically significant at less than the level of significance (1%). The obtained estimate of disease intensity for the Malnad region irrespective of sample weeks and sample years was 2.38. The rate ratio for the same was estimated by taking its antilog as 10.80, indicating that disease intensity in the Malnad region is 10.80-times more than the control group, i.e., the Maidan region. Similarly, the rate ratio for the coastal region was estimated at 8.94, reflecting that the disease intensity in the coastal region was 8.94-times more than the control group (Maidan region).

Concerning sample weeks, irrespective of regions and years, the PDI indicated by the rate ratio for the sixth week reflected that disease intensity was lower by 91% compared to the 18th week (control week). With the progression in the sample weeks, the likelihood of disease intensity increased. For instance, the disease intensity in the 10th week was 72% lower than the 18th week, while it was 1% lower in the 17th week compared to the 18th week (control). Year element has a less significant influence on the response variable with a rate ratio of 1.05, indicating that PDI was 5% higher in 2019 than in 2018.

The negative binomial regression model is additive in the log scale, and the additive model fits reasonably well for the data considered in the study. The predicted values of PDI across sample weeks for all the three regions and differences between Malnad and Maidan; coastal and Maidan are presented in Table 4. The regional differential/gradient of tenfold translates the PDI to 6.62 in the sixth week and reaches 77.84 in the 18th week. Thus, it could be inferred that the absolute effect of regions measured in the original scale increases with sampling weeks. The same analogy holds for the coastal region wherein the regional differential/gradient of eightfold translates PDI to 5.45 in the sixth week and reaches 64.06 in the 18th week.

### 3.4. Dynamics of FRD over Space

Spatial interpolated maps generated using ordinary kriging (OK) showed the spatial aggregation pattern of disease intensity over weeks (*n* = 18) of observation in the varied regions (*n* = 3). Among the studied areas, FRD was widely distributed and spatially aggregated in the Malnad and coastal regions compared to the Maidan region, which had random occurrence of FRD.

#### Spatial Interpolation through the OK Approach

The results of OK are depicted through color-coded maps where the darker color (red) represents a higher disease intensity of FRD and lighter color (green) indicates a lower disease variable. Of the total interpolated surface at the Malnad region during 2018 and 2019, 2.5% and 6% showed higher disease intensity ranged from 61–70%, respectively. 

The disease intensity up to 51–60%, 41–50% and 31–40% was observed in 22.3%, 25% and 7.5% of the total surface, respectively, indicating spatial aggregation of disease during 2018. Subsequently, around 60%, 12% and 3% were observed in 2019 (Figure 5A). This pattern of spatial clusters evidenced that the disease intensity was varied among sampling sites of the region with varying disease proportions and identified risk areas highly vulnerable to the disease spread and occurrence.

From the appearance of disease until the end of the period during both seasons, the disease intensity displayed an aggregated pattern of disease occurrence in the coastal region. The maximum surface area during 2018 (about 90%) exhibited disease intensity between 41% and 50%, and the remaining (10%) areas witnessed disease proportion between 51% and 60% representing potential risk areas (Figure 5B). Throughout the season of 2019, 50% of the interpolated surface area depicted the disease intensity of >51% and 41%, respectively.

When looking at the distribution of disease intensity overall plots, the Maidan region indicated an unequal or scattered pattern. The disease intensity ranged from 21 to 30 % in 2018 than 2019, which was relatively lesser than other studied regions. Surprisingly, around four sampling sites in the region comprised no disease intensity (40% interpolated areas), which is non-obvious compared to other regions depicting less disease-prone or risk areas. 

In 2018, 11–20% of disease intensity was observed in 55% of the total interpolated surface, and the remaining 5% of areas consisted of a disease proportion of about 21–30% (Figure 5C). The disease intensity during 2019 varied between 11% and 20%, which shared at 60% of the total surface, and only 1% had disease intensity ranged from 21–30%. It seems that general trend was maintained across the regions in both years.

### 3.5. Spatial Autocorrelation

The Moran’s *I* statistical test on the PDI for 18 standard weeks during 2018 and 2019 at the observational sampling points in combined regional data were depicted in Table 5. Moran’s *I* spatial autocorrelation confirmed the presence of a significant (*p* < 0.05) spatial clustering pattern among PDI of FRD from the sixth week onwards. At all sampling sites (*n* = 27), up to the fifth standard week, the Moran’s I statistic value was zero, indicating no distinct clusters observed across the studied areas (Table 5), which is also reflected in interpolation results. After the sixth week onwards, a strong spatial dependency among 27 sampling sites was witnessed and significantly exhibited distinct spatial clusters (*p* < 0.05). Furthermore, the spatial distribution pattern depicted the similar clusters indicating that the same cluster category was bounded on another cluster category with Moran’s *I* statistic close to 1.

### 3.6. Influence of Climatic Variables on Fruit Rot Epidemics

#### 3.6.1. Correlations between Fruit Rot Intensity and Climatic Determinants

For all the environment variables (*n* = 7), a strong relationship between points were observed in all the plots. Bivariate correlation analysis revealed a significant (*p* > 0.01) correlation between environmental variables and FRD intensity across the sampling sites of Karnataka.

The results from the combined dataset, the correlation between T_max_ and T_min_ (r = 0.80, *p* = 0.01), RH (r = 0.88, *p* = 0.01), WS (r = 0.79, *p* = 0.01), TRF (r = 0.88, *p* = 0.01) and FRD occurrence was strong positive and significant (*p* > 0.01) but correlation of RD with disease intensity was slightly positive (*p* > 0.01) (Figure 6). The correlation results output clearly indicated that FRD has strong relation with temperature, relative humidity and rainfall across the sampling sites. This variation in prevailed environmental situations and microclimate within or between gardens suggest significance of the weather parameters in FRD epidemics at various sampling sites in Karnataka.

Furthermore, the plot indicating the maximal PDI of each of the 27 sites against integrated climatic variables over the 18 weeks period inferred in Figure 7. This clearly deciphered and supported the results of correlation analysis as well as allowed us to understand the impact of climatic indicators against PDI across the sampling sites.

#### 3.6.2. Multiple Linear Regression Model

Table 6 represents the parameter estimates to understand the possible associations between occurrences of FRD and climatic covariates along with corresponding confidence interval (CI). Based on combined dataset (2018 and 2019), there was considerable association was noticed between climatic variables and PDI across the studied areas. The significant association between T_max_, RH1 and TRF with response variable (PDI) was determined. In these results *p*-value (significance) for T_max_, RH1 and TRF is ≤0.001, which is less than significance level of 0.05. 

This indicated that the association between T_max_, RH1 and TRF with PDI statistically significant. The coefficients for the predictors T_max_, RH1 and TRF are 1.731, 1.330 and 0.541, respectively. This indicated that the FRD intensity increases by 1.731, 1.330 and 0.541 for every 1 unit increase in T_max_, RH1 and TRF. The sign of the coefficient is positive, which revealed that as T_max_, RH1 and TRF increased, FRD intensity also increased (Table 6 and Appendix A). The remaining predictors, such as T_min_, RH2, WS and RD, were found slightly associated with response variable as level of significance is more than 0.05. This represented that the occurrence of FRD is independent on T_min_, RH2, WS and RD factors.

The overall results from regression model analysis demonstrated that, maximum temperature, relative humidity and rainfall pattern were found to be key determining factors in prevalence and spread of the FRD in arecanut plantations over studied sites of Karnataka, India.

## 4. Discussion

### 4.1. Spatial Pattern Analysis of FRD

The studied regions were varied by climatic profiles, altitudinal characteristics and initial inoculum of the pathogen, which decides the distribution of fruit rot epidemic across the regions. The disease pattern displayed a gradient across the regions as confirmed by ordinary kriging (OK) and spatial dependency, which represented the nature of disease distribution at sampling sites throughout investigated areas. 

The spatial clustering was varied throughout the regions as evidenced by spatial autocorrelation analysis, which had significant spatial clusters between weeks at all the sampling sites (*n* = 27). The spatial analysis and dependency exhibited an aggregated pattern of infections by the pathogen at Malnad and coastal regions; however, Maidan regions had observed random patterns. The findings of our study are corroborated with similar results on the spatio-temporal patterns of bud rot on coconut, which observed in multiple sites and spread across the studied area with regular and contagious or cluster-distribution patterns [44].

Ordinary kriging (OK) generated identical FRD prediction surfaces at Malnad and coastal regions but Maidan region had lesser predicted surface area and disease demonstrated relatively higher spatial correlation indicating spatial clusters. Due to presence of spatial dependency, the kriging approach was considered the most suitable surface interpolation technique than IDW due to distance-based outcomes; hence, OK generated maps were recommended in this study [45,46,47]. 

The relative geographical distribution of FRD showed the potential risk areas in the studied regions of Karnataka. The surface interpolated maps demonstrated the higher potential risk areas mainly concentrated in high altitude areas on Western-Ghats (Malnad tracts), Plateau and hilly areas and Seashore (coastal belt). By considering the results from OK demonstrated that, the topographical effects highly influenced disease distribution. A decade ago, disease was increasing to non-traditional plain areas (Maidan region) as revealed in OK surface maps, including arecanut plantations established along the riverside (Figure 8).

The variations in the aggregation of spatial effects between sampling sites at Malnad and coastal regions might be credited to the difference in the age of the palms; hence, gardens at Malnad and coastal regions were established before long back (>25 years older) than other Maidan regions (recently established) would be highly vulnerable to attack by the pathogen (*P. meadii*). Over the years, the initial disease intensity that existed might have disseminated throughout the arecanut plantations; however, the random pattern of disease distribution was witnessed at Maidan region. 

This idea is further supported by data, which showed that in the plantations >20 years old and well established, 16 samples (88%) out of 18 tested positive for *P. megakarya* [48]. The intra-plot environment, especially the presence of other plantation crops, such as cocoa, black pepper and banana, as potential intercrops between the rows of arecanut at the Malnad and coastal region influenced the inoculum buildup and spread within the garden due to increased relative humidity.

Deviations in disease patterns over space were attributed to rainfall patterns, congenial environmental situations, development of microclimate within gardens, dispersal of the inoculum within palms, movement the infectious spores between-palms, fields and neighboring gardens; management practices. It was evident that, the disease was highly correlated with temperature, relative humidity and rainfall pattern at Malnad and coastal regions. It was hypothesized that the presence of initial inoculum is the key factor for the spatio-temporal development of FRD at the plantation level [26]. The temperature has significantly influenced the occurrence of *P. megakarya* on cocoa plantations at a lower temperature between 23 and 26 °C [49,50], and the survival ability of *P. megakarya* showed that the soil is not only medium also the existence of intra-tree dispersal might be credited [51].

When compared to between and within the years, the location of infection clusters was consistent in coastal and Malnad regions. In contrast, it was not consistent in the Maidan region. When looking at the data for the Maidan region, there did not have particular area that was consistently infected (random spatial effect was witnessed). This was reflected in the microclimate data, which did not show a plot having optimal microclimatic conditions favoring pathogen development. 

In Malnad and coastal regions, microclimate was most conducive for the disease development in the plot where infections were consistently noticed. These areas receive more shade in the morning than other areas, which could induce dew on the surface of arecanut bunches for a prolonged period, facilitating the germination of spores of pathogen. These results were consistent with the reports demonstrated the persistent dew on the pod surface for up to 3 h after sunrise is ideal for black pod rot disease development [52]. Primary inoculum initiates disease and influences the development of epidemic thought-out the year was demonstrated on cocoa plantations [53].

### 4.2. Temporal Dynamics of FRD

It was expected that, over time, an increase in infections to larger areas with disease intensity was observed at Malnad and coastal regions. However, this was not the case in the Maidan region, where the disease was observed on a limited scale. Over time, disease progression across regions revealed that secondary infections are most often derived from primary ones than other infections. However, most infected palms generated secondary or primary inoculum, contributing to them being infected within the same year. The lack of increase of the disease over time is thus because secondary inoculum mainly causes self-infection but fails to disseminate between palms in the garden. This supports the hypothesis of Ten Hoopen et al., 2009 [26], demonstrating that the primary inoculum is the prime factor that favors the development of an epidemic over time at the plantation level.

Although variability in the distribution of rainfall and quantity could explain differences in disease intensity between years, it was remained relatively low in plots (*n* = 9) of the Maidan region. Two potential explanations could be attributed, weekly phytosanitation and fungicide treatments, to consider when interpreting the results. In Malnad and coastal regions, fungicidal applications were taken on a preventive basis. In both cases, the subsequent year showed a decrease in disease intensity; however, this decrease was quite substantial in the case of the Maidan region because of late occurrence of rainfall. However, rainfall is the prime explanatory variable for temporal dynamics of disease and is mainly linked to the number of infected palms within a garden [25,49,54].

As discussed above, hilly, Western-Ghats, Seashore areas had considerable variation in altitude, which characterized a microclimate with lower temperature, continuous erratic rainfall and higher humidity favors FRD development and spread. In arecanut-growing areas, these kinds of terrains are aggregated spatially, which resulted spatial patterns in disease intensity. The aggregations of FRD at a regional scale across districts and taluks were unequal due to the nature of disease prevalence. This was partially confirmed by the observation of no strong spatial autocorrelation at the Maidan region. The lack of spatial dependence at Maidan region and the strong spatial correlation was at Malnad and coastal over weeks across regions suggested that the presence of initial inoculum and microclimate prevailed in gardens (which could be from infected bunches, treetop and various pathogen structures surviving in the soil) might have played a significant role in the fruit rot epidemics on arecanut in major producing areas of Karnataka.

### 4.3. Implications in FRD Management

The pattern of spatio-temporal distribution of fruit rot on arecanut has very significant implications. Due to aggregated patterns of fruit rot at a regional scale, the disease monitoring attention should be focused on altitude and weather conditions. The distribution of disease and relatively stable high probable risk areas over the years would help us to design suitable disease management approaches. 

High-risk areas are usually Western-Ghats (Malnad), hilly and mountainous areas (coastal), often located at the junction of several districts or states. Therefore, we should promote a professional pest control system for large-scale disease prevention. Most importantly the farmers and extension agents could use the spatial distribution maps generated in our study in devising precautionary measures and formulating management stratums to regulate the further dissemination of disease to neighboring districts or states. The highly vulnerable areas to disease were identified in our study, indicating that farmers in those areas should resort management approaches to curtail the disease effectively.

## 5. Conclusions

We demonstrated the spatial and temporal dynamics of FRD at various sampling sites across different regions (Malnad, coastal and Maidan) of Karnataka. The present study constitutes the first documentation of spatial and temporal dynamics by employing geostatistical and GLM analysis to generate gradient spatio-temporal maps across the regions. The spatial pattern and autocorrelation on disease intensity at twenty-seven sampling plots across regions displayed random and aggregated effects over the evaluated seasons. 

Further, temporal dynamics revealed that the early initiation of the epidemic was witnessed at the Malnad and coastal regions in contrary to the Maidan tracts where the occurrence was delayed up to the end of the season. The regional occurrence of FRD was greatly influenced by the average temperature (25–30 °C), relative humidity (>90%) and erratic rainfall (>1000 mm) as evident from the correlation and regression analysis. Furthermore, the results indicated that arecanut plantations with the presence of initial inoculum exhibited the maximum disease proportion over space and time as revealed in the temporal analysis. However, more attention should be focused on the reduction of initial inoculum in the gardens. The actual surface interpolated maps generated in this study would help to develop suitable management approaches, which rely on the reduction of further spread of FRD to neighboring areas. The epidemiological information on FRD in this investigation will be useful in developing forecast models as well as a decision support system as an aid for the design and implementation of control strategies for the disease.

## Figures and Tables

**Figure 1 jof-08-00745-f001:**
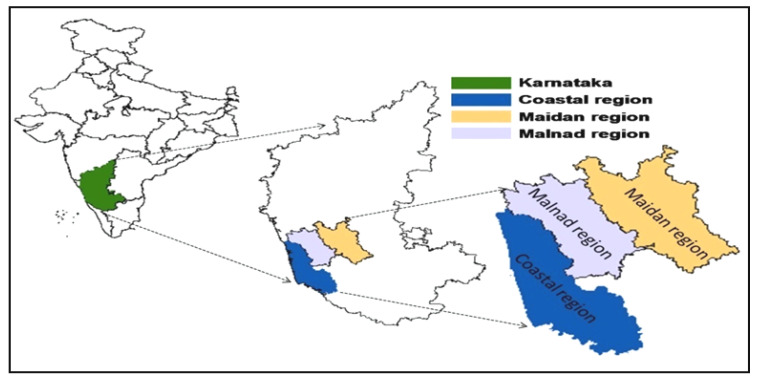
Administrative map of India (Left), Karnataka state (Middle) and experimental regions to study the spatio-temporal distribution of fruit rot disease in Karnataka (Right).

**Figure 2 jof-08-00745-f002:**
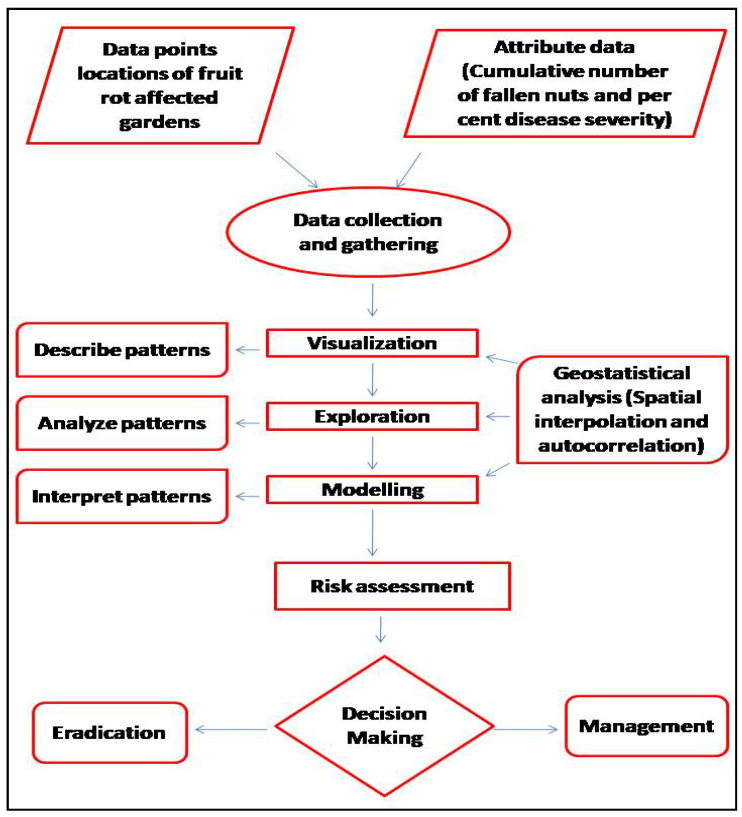
Conceptual frameworks to understand the spatial and temporal pattern analysis of FRD in Karnataka.

**Figure 3 jof-08-00745-f003:**
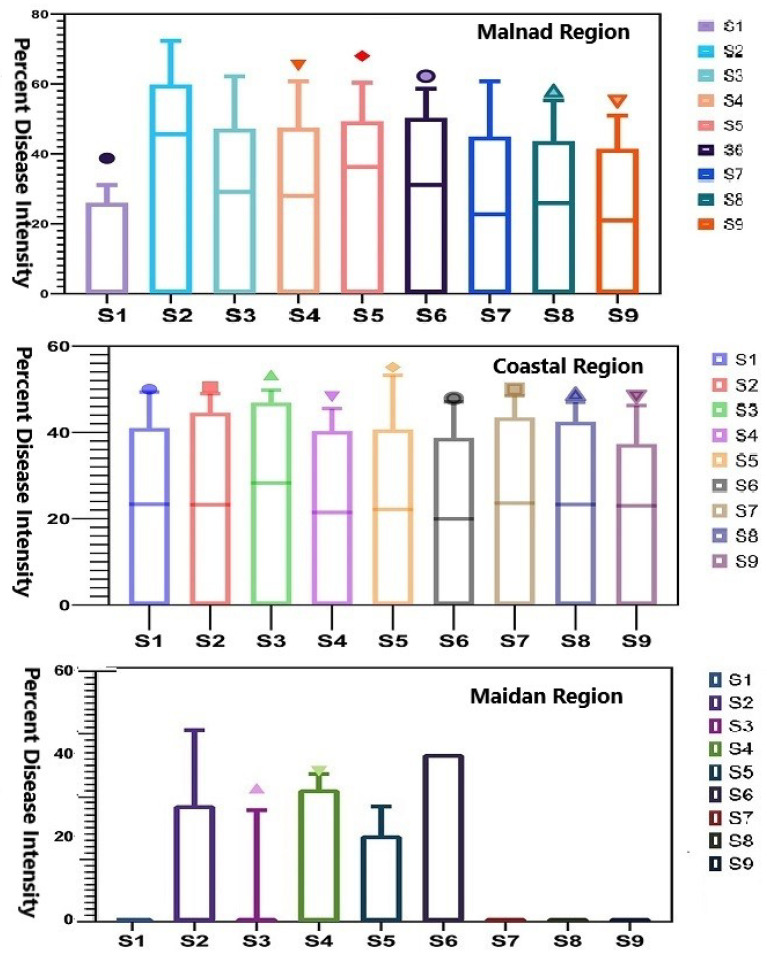
Box and whisker plot representing intensity (PDI) of fruit rot across different regions of Karnataka. The middle bar = median, box = inter-quartile range (25th–75th percentile) and whiskers (error bars) above and below the box represents the 90th and 10th percentiles.

**Figure 4 jof-08-00745-f004:**
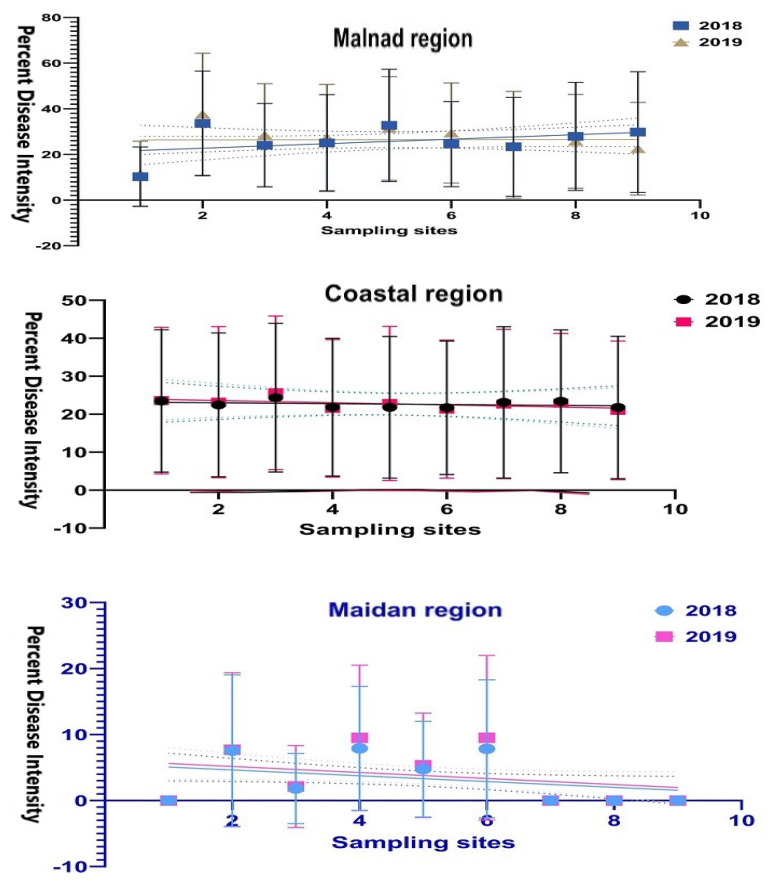
The temporal progress of arecanut fruit rot PDI during two consecutive years at sampling sites in each region. Each point was the mean of 18 weeks of PDI data at different sampling sites (*n* = 9) collected during 2018 and 2019. Graphs represented the mean, range and standard deviation of PDI, and the vertical bar represents the standard error of means.

**Figure 5 jof-08-00745-f005:**
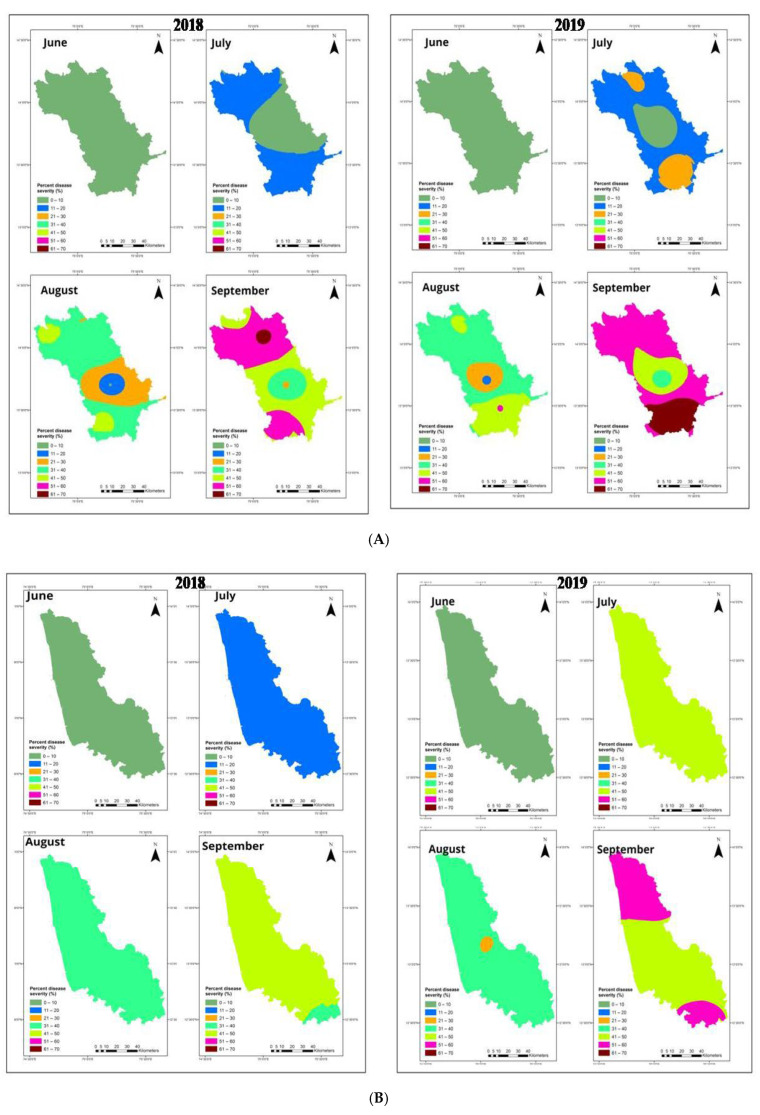
Interpolated maps represent disease intensity of FRD using ordinary kriging approach at Malnad (**A**), coastal (**B**) and Maidan (**C**) regions of Karnataka during 2018 and 2019.

**Figure 6 jof-08-00745-f006:**
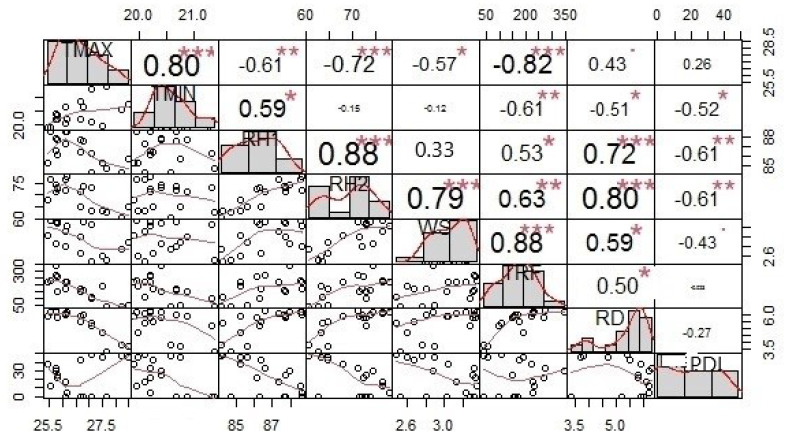
Correlation analysis between FRD intensity and climatic conditions across the 27 sampling sites s of Karnataka. T_max_ (Maximum Temperature), T_min_ (Minimum Temperature), RH1 (Morning Relative Humidity), RH2 (Evening Relative Humidity), WS (Wind Speed), TRF (Total Rainfall), RD (Number of Rainy Days) and PDI (Percentage Disease Intensity). Significance codes: *** —0.001, ** —0.01, * —0.05.

**Figure 7 jof-08-00745-f007:**
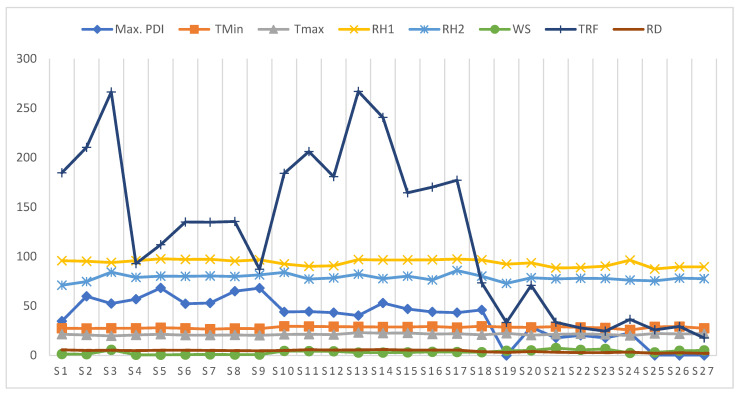
Graph representing the maximum PDI and integrated climatic variables across the 27 sampling sites and deciphered the clustering of level of PDI with climatic indicators.

**Figure 8 jof-08-00745-f008:**
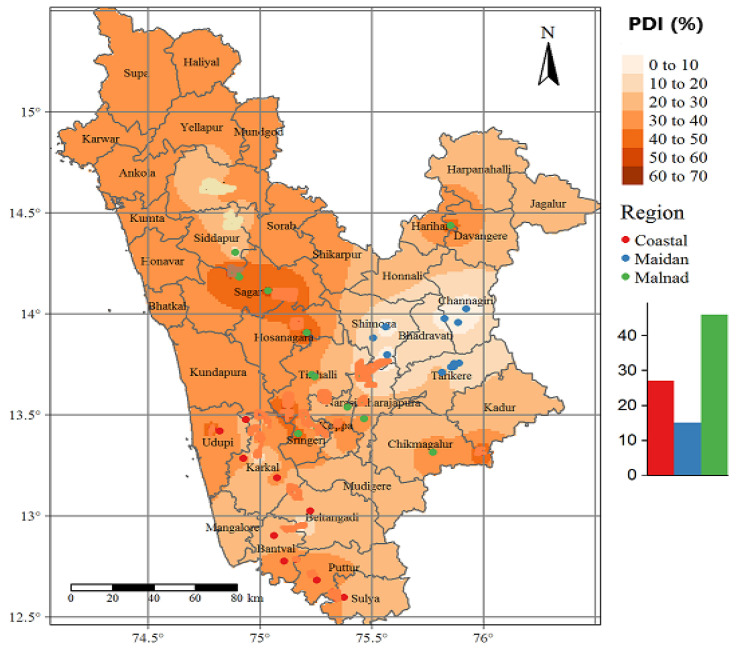
Topographical map representing three regions with 27 sampling sites located in Karnataka.

**Table 1 jof-08-00745-t001:** The climatic diversity and ecological profile of the three regions of Karnataka considered during this study.

Characteristics	Malnad	Coastal	Maidan
Agro-ecological zones	The Western Ghats and Hot-Humid region (19.2)	Coastal Plain, Hot Humid Region (19.2)	Sub-Humid plain region (19.1)
Agro-climatic Region	Ghat region	West Coast Plains	Southern plateau
Agro-climatic Zone	Hilly (Malnad) zone	Coastal Zone	Northern transition zone
Altitude (m)	1119 m	69 m	631 m
Latitude (Boundary)	13°55′20.10″ N	12°50′04.02″ N	13°18′44.72″ N
Longitude (Boundary)	75°34′14.73″ E	75°14′54.92″ E	75°46′13.10″ E
Annual rainfall (mm) *	1812.5	2559.4	906.3
Average temperature (°C) *	25.6 °C	27.6 °C	29.8 °C
Soil type	Laterite clayey soils Red Laterite soils	Red Laterite soils Sandy loam soils	Red clay soils

* Average values of two consecutive years 2018 and 2019.

**Table 2 jof-08-00745-t002:** Description of the disease rating scale used to assess the PDI of FRD in arecanut [31].

Scale	Description
1	1–10% nut fall from bunches
2	11–25% nut drop
3	26–50% nut drop
4	51–75% nut drop + spread of the disease to bunch stalk
5	76–100% nut drop + spread of the disease to the main stalk of the bunch
6	Crown death

**Table 3 jof-08-00745-t003:** Summary and estimates of GLM negative binomial regression model with log link.

Covariates	Estimates	Standard Error	Rate Ratio	z Value	Pr (>|z|)	Significance
Intercept	1.971	0.136	7.179	14.533	2 × 10^−16^	***
D1	2.384	0.088	10.843	26.951	2 × 10^−16^	***
D2	2.189	0.089	8.924	24.701	2 × 10^−16^	***
D3	–2.465	0.186	0.085	–13.229	2 × 10^−16^	***
D4	–1.838	0.179	0.159	–10.291	2 × 10^−16^	***
D5	–1.592	0.176	0.204	–9.026	2 × 10^−16^	***
D6	–1.407	0.175	0.245	–8.046	8.58 × 10^−16^	***
D7	–1.276	0.174	0.279	–7.34	2.14 × 10^−16^	***
D8	–0.986	0.172	0.373	–5.73	1.01 × 10^−13^	***
D9	–0.729	0.171	0.482	–4.269	1.97 × 10^−5^	***
D10	–0.398	0.169	0.672	–2.35	0.0188	*
D11	–0.293	0.169	0.746	–1.733	0.0831	.
D12	–0.206	0.169	0.813	–1.223	0.2212	
D13	–0.131	0.169	0.878	–0.775	0.4385	
D14	–0.006	0.168	0.994	–0.036	0.9716	
D15	0.050	0.069	1.052	0.728	0.4668	

Significance codes: ***—0.001, *—0.05, ‘.’—0.1 and no marking—1. (Dispersion parameter for Negative Binomial (1.4024) family taken to be 1). Null deviance: 1876.83 on 701 degrees of freedom. Residual deviance: 933.88 on 686 degrees of freedom. AIC: 5216, Number of Fisher Scoring iterations: 1.

**Table 4 jof-08-00745-t004:** Predicted values of PDI for arecanut growing regions across the sampled weeks.

Regions/Weeks	6th	7th	8th	9th	10th	11th	12th	13th	14th	15th	16th	17th	18th
**Malnad region**	6.62	12.39	15.85	19.07	21.72	29.04	37.55	52.28	58.08	63.32	68.31	77.38	77.84
**Coastal region**	5.45	10.19	13.04	15.69	17.88	23.90	30.91	43.03	47.80	52.12	56.22	63.68	64.06
**Maidan region**	0.61	1.14	1.46	1.76	2.00	2.68	3.46	4.82	5.36	5.84	6.30	7.14	7.18

**Table 5 jof-08-00745-t005:** Summary of the spatial dependency on intensity of FRD of arecanut in Karnataka, India.

Standard Week	2018		2019	
	Moran’s *I* Statistic	*p*-Value (0.05)	Moran’s *I* Statistic	*p*-Value (0.05)
**W1**	0.00	0.00	0.00	0.00
**W2**	0.00	0.00	0.00	0.00
**W3**	0.00	0.00	0.00	0.00
**W4**	0.00	0.00	0.00	0.00
**W5**	0.00	0.00	0.00	0.00
**W6**	0.151	0.160	−0.005	0.820
**W7**	0.591	0.06	0.460	0.022
**W8**	0.650	0.01	0.562	0.014
**W9**	0.656	0.001	0.559	0.001
**W10**	0.639	0.02	0.550	0.0027
**W11**	0.682	0.001	0.516	0.0074
**W12**	0.657	0.005	0.498	0.001
**W13**	0.620	0.0001	0.522	0.0057
**W14**	0.600	0.006	0.524	0.005
**W15**	0.630	0.008	0.552	0.002
**W16**	0.660	0.0041	0.584	0.008
**W17**	0.700	0.0036	0.576	0.009
**W18**	0.715	0.0075	0.623	0.002

**Table 6 jof-08-00745-t006:** The estimated effects of climatic indicators on FRD analyzed through multiple linear regression model.

Covariates	β	*t*-Value	Sig.	95% Confidence Interval (CI)	
				Lower Limit	Upper Limit
(Constant)	168.51	3.885	0.001 ***	72.241	210.316
T_max_	1.731	4.514	0.001 ***	0.795	6.235
T_min_	1.006	8.213	0.030 **	−6.473	20.852
RH1	1.330	4.418	0.001 *	−18.150	2.421
RH2	1.064	1.791	0.013 **	−3.381	1.856
WS	0.452	0.435	0.005 *	−1.005	4.526
TRF	0.541	−0.673	0.001 **	−2.583	1.627
RD	−0.604	−2.182	0.117 *	−2.851	0.901
Region	2.065	1.935	0.001 ***	0.126	3.172

T_max_ (Maximum Temperature), T_min_ (Minimum Temperature), RH1 (Morning Relative Humidity). RH2 (Evening Relative Humidity), WS (Wind Speed), TRF (Total Rainfall), RD (Number of Rainy Days). Sig. indicates the level of significance at (*p =* 0.05); and β represents the estimated coefficients of variables. Significance codes: ***—0.001, **—0.01 *—0.05.

## Data Availability

The data presented in this study are available on request from the corresponding author.

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
