# Peer review of "Exploring the Impact of Climatic Variables on Arecanut Fruit Rot Epidemic by Understanding the Disease Dynamics in Relation to Space and Time"

_jof, 2022, doi:10.3390/jof8070745_

Round 1
Reviewer 1 Report
There is a real potential: the dataset is rich and the analyses are interesting. However, the authors have to get to the points, make choice and reduce redundancies. Some analyses and figures are redundant and some results are missing. I did suggestions to help to clarify points, reduce text and add missing information.
I think this is too much to only ask for 'minor' revision.
The paper deals with fruit rot disease of arecanut, caused by Phytophtora meadii in India. Its aim is to decipher the role of climatic factors on the spread of the disease, by analyzing a rich dataset of observations. Indeed, disease occurrence was monitored through 27 sites over 3 regions, along 18 weeks during the monsoon period, for 2 years. The analyses performed are robust and of interest. Spatial and time interpolations are proposed as well as a study of the effect of climatic variables. Overall, this paper is of high interest to progress in the understanding of the epidemiology of the disease. It proposes interesting analyses of the data and advances the current knowledge. In addition, it may be helpful to propose better surveillance systems.
However, there are quite many improvements to be done before publication. Especially, some information is missing to help the understanding and to support the conclusions (for example: location of the 27 sites on the maps, topography of the 3 regions, summary of climatic characteristics of the sites and regions; alternative analyses). The authors have to choose the figures to plot and avoid redundancy. But the main troubling point concern the inconsistencies between the correlation and the regression analyses for climatic variables: either it was difficult to understand or they are errors. This has to be fixed.
Also, English syntax has to be improved. Specific points are listed above:
Introduction
For people not familiar with Arecanut production, information is missing. For example, what is the period of harvest of the nuts in these regions ? Are sane Nuts fall during June-sept or fallen nuts are only the one infected? It is not clear if Arecanut fruit rot is always due to Phytophtora meadii only or if fruit rot can be caused by other fungi. The species, namely Phytophtora meadii, is even not mentioned in the article summary. This is missing.
Lines 66-70 : it is not clear what is general information of the impact of the disease (that should be written in present) and what are specific events to illustrate. For example, of losses, year and site have to be mentioned.
Lines 71-96 : the two paragraphs have to be reorganized. There is i) information on the presence of the disease in different regions and its spread and on the epidemiology of the pathogen, but they are mixed. In addition, information on the provinces and states is not understandable by non-natives of India. A simplification would be appreciated. A map could also be added.
Line 86: FRD is not defined the first time used.
M&M
The experimental design should be described at the beginning of M&M, not in introduction of 2.6. They are few explanations missing:
- are the 50 palms observed the same across the weeks and for the 2 years?
- how do you know if fallen nuts are new ones, not observed the week before? Do you remove the fallen nuts from the ground each week?
- in the calculation of PDI, the ‘Maximum rating’ is the one observed across all the garden studied and all weeks of observations, or is 6 the maximum of the scale used for observations?
Through the manuscript, disease ‘severity’ is used. Please always use the same words: disease intensity and
Percent disease intensity (PDI) for clarity. They are such incoherencies through the figure legends and axes names. In equation (3), x0 is not explained. What does ‘z is the variable of interest at locations xi and x0‘ mean?
The organization of the sections 2.5 and 2.6 has to be modified. There is 2.6.1 but no 2.6.2 and part f the information in 2.6 introduction is already in 2.5. Both sections can be merged. Section 2.6.1 could be shortened to avoid repetitions, especially between text and equation.
In section 2.8, reorganize to avoid repetitions: the software used (ArcGIS 10.3) is cited 3 times and the loc transformation twice.
In section 2.9, line 296: what is district ? site, region ?
line 297: if the average PDI is calculated on the 3 regions of study, please write it clearly; if not explain more.
Section 2.10:
Lines 307-312 : this information is not understandable by non-natives of India. Please clarify.
Please add the model of regression analysis and the R package used. Explain the confidence intervals that are given as outputs of the analysis. This is not obvious.
Results
Sections 3.1 and 3.2
To first present the PDI across sites, I would propose to only show the maximal PDI of each site each year. This would allow to avoid a mean along time that is not very meaningful.
For Maidan area, the map in Fig 6 pointed a zone with lower infection than in the rest of the area. Is it where site 1 is located? In the text, it is not mentioned that S1 has lower infection than other sites (only S2 is said higher than others), but it seems so in Fig 3. Looking at the maximal PDI in each site and year would allow to compare the sites in a simplest way. In addition, it would be helpful to locate the 9 sites if study in each of the 3 regions.
Results presented in the 2 plots (A and B) of Fig 3 are partially redundant and Fig 3 B is redundant with Fig 4 B. They are very difficult to read because not same colors nor scale were used across regions.
I recommend less figures but bigger so the information can be properly seen: for example, the dynamics in Fig 4B is not visible: larger plots are needed.
I think Fig 4B, done for each year would summarize all the information of Fig 3 and 4 : 2 column, one for each year, and extended the X-axis to properly see the trends over weeks.
The statement that the 2 years are different is not convincing. Indeed, in Fig 4A, the different statistics plotted
are very similar (means, ranges and errors…).
Line 401: replace ‘Time’ by ‘Year’ because ‘Time’ could also refer to weeks.
Line 416: ’77.14’ should be ’77.84’ according to the Table 4. It is not necessary to give the value for week 10th : 1st and last are enough.
Section 3.4 : the utility of this section is not obvious. Figure 5 is redundant to Figure 4 B from which same results can be inferred (if 2 plots are drawn, one for each year). It does not provide more information about the disease and the regions. I would remove this section to simplify the paper which already contains many different analyses.
Section 3.5.1: the text for Coastal results is not in adequation with the figure 6 B. There might be a problem with the legend and colors. For example in 2019, according to the figure, 50% is over 51 PDI and 50% over 41 not over 31 and 21 as written in lines 468.
For Maidan region, there is more area in orange (21-30) in 2018 than in 2019, contrary to what written on lines 472-473. What those lines mean is not clear.
The end of the paragraph, lines 481-483, is not convincing. It seems that general trend across the region are maintained the 2 years.
Section 3.6: the text is difficult to follow because it seems not always related to the table 5 mentioned which show results for the 2 years (not regions).
Lines 497-500: from where this information comes ?
Lines 497 to 504 deal with the regions. I guess these results come from the section before.
Finally, the adding value of calculating the Moran’s I test is not clear from this section. It does not bring any more information than the interpolation approach ; or maybe only statistical results. So I would recommend to suppress it or to include the results to support the results of the interpolation.
Section 3.7.1
Lines 510-515: what ‘differences’ mean here? I guess it is ‘relationships’ or ‘correlations’ to be used instead. According to Figure 7, the correlations were not always significant, contrarily to what written. The 2 first sentences are very redundant with the 3rd one : this could be shortened and modulated because some correlations are no significant: generalization is not possible.
The negative links seen between RH and PDI is really surprising. Usually fungal disease develops with high humidity, which is in adequation with the fact the disease develops after monsoon rains. Unfortunately, no information was given about the biology of the fungus in introduction and nothing is said on this negative relationship.
I would strongly suggest to do these analyses merging the 3 regions. This would allow to enlarge the range of the variables and increase the number of points and may help to get stronger correlations. It would also be interesting to plot the maximal PDI of each of the 27 sites against integrated climatic variables over the 18 weeks period : this would allow deciphering the climatic indicators that could explain the clustering of some areas with lower disease.
Section 3.7.2
Why not include a region effect in the regression model and try to extract from the whole dataset the climatic variables that influence most the disease? It would be more powerful and avoid to get contrary results from the different regions.
The interpretation of the table is difficult to follow: the model was not presented in M&M and the outputs are not explained. Which variable gives the significance of the effect ? if it is ‘Sig.’ then why is it negative sometimes
? and very few effects prove to be significant…
In general, this section is very disappointing since the results are not in adequation with the section before: very often relations have opposite sign to the correlations… It is weird and not explained. The conclusion in lines 551- 553 is not inline with Figure 7.
So, it seems that the objectives of this section are not clear : what it brings in addition to section 3.7.1?
Discussion
The link between disease occurrence and topography is interesting and it’s a pity it was not put in light before. Maybe one easy way would to provide a topographical map of the 3 regions with the 27 sites located on it. This would at least give an idea of the location of the sites and give the opportunity to see the link proposed in lines 582-584.
In accordance with the discussion lines 643-645, it appears necessary to better illustrate the climate diversity at the different sites and the 3 regions. Highlight should be made at the beginning of the paper, on the fact there is lower rainfall at Maidan region.
Thee definition of ‘Wester-Ghats’ have to be given for non-native of India.
Considering the importance of dew in the morning, it would be interesting to calculate leaf wetness. The same way, taking into account an indicator of inoculum pressure of the sites in the regression model could allow to get more accurate results in a general model including all sites.
Figures
Figure 3: A and B are partly redundant and difficult to read; In Fig 3B X-axis is problematic: sites are called with letters. Replace severity by intensity
Figure 4: replace severity by intensity. Use same scales of axis when multi-plots to ease the comparisons. Table 4 : remove the lines for Difference 1 and 2 that are not useful.
Figure 6: the colors used for the 1st and 4th classes are far too similar. It’s difficult to differentiate them. Would
help for the whole article to position the 9 sites (points + S1 to S9) studied in each region. Figure 7 and Table 6: add the definition of the letters for climatic variables following the titles Table 6 : why the model is said ‘spatio-temporal’ ?
Add the signification of the abbreviations of the climatic variables. use same coding as in the text (M&M, Results and Figure 7): RH1 or RH-I…
Explain the columns of the table : Sig. for example. If it is ‘significance’, why could it be negative ?
English syntax has to be thoroughly revised along the text. Many sentences are not correct.
Author Response
Author’s Response to Reviewer 1 Comments
Dear reviewer
Thank you very much and we appreciate the time and effort that you dedicated to provide feedback on our manuscript and grateful for the insightful, constructive comments and valuable improvements to our paper. We have incorporated all of the suggestions and those changes are highlighted (in track changes mode) within the manuscript. Please see below, in red, for a point-to-point response to the reviewer’s comments and concerns.
Specific comments
Introduction
Point 1: For people not familiar with Arecanut production, information is missing. For example, what is the period of harvest of the nuts in these regions?
Response 1: This suggestion was accepted and modified accordingly as revealed in introduction section.
Point 2: Are same Nuts fall during June-sept or fallen nuts are only the one infected? It is not clear if Arecanut fruit rot is always due to Phytophthora meadii only or if fruit rot can be caused by other fungi. The species, namely Phytophthora meadii, is even not mentioned in the article summary. This is missing.
Response 2: Nut fall during June-September happed due to Phytophthora infection and is the only agent which causes fruit rot disease in arecanut (no other fungal association have been reported). Species name P. meadii has been mentioned in summary as per the suggestion.
Point 3: Lines 66-70: it is not clear what is general information of the impact of the disease (that should be written in present) and what are specific events to illustrate. For example, of losses, year and site have to be mentioned.
Response 3: Sentence has been changed and illustrated the impact of disease by including losses caused, year and site.
Point 4: Lines 71-96: the two paragraphs have to be reorganized. There is i) information on the presence of the disease in different regions and its spread and on the epidemiology of the pathogen, but they are mixed. In addition, information on the provinces and states is not understandable by non-natives of India. A simplification would be appreciated.
Response 4: Two paragraphs were reorganized and simplified as per the suggestion (line: 71-96).
Point 5: Line 86: FRD is not defined the first time used.
Response 5: Expanded and changed accordingly
Materials and Method
Point 1: are the 50 palms observed the same across the weeks and for the 2 years?
Response 1: During the study, randomly 50 palms were selected and tagged, subsequently the same palms were monitored across weeks for two years.
Point 2: how do you know if fallen nuts are new ones, not observed the week before? Do you remove the fallen nuts from the ground each week?
Response 2: While recording the observations across 18 weeks from June-September, infected fallen nuts were removed and destroyed after each data recording.
Point 3: in the calculation of PDI, the ‘Maximum rating’ is the one observed across all the garden studied and all weeks of observations, or is 6 the maximum of the scale used for observations?
Response 3: In PDI calculation, 6 is the maximum scale used for rating (0-6 scale)
Point 4: Through the manuscript, disease ‘severity’ is used. Please always use the same words: disease intensity and Percent disease intensity (PDI) for clarity. They are such incoherencies through the figure legends and axes names. In equation (3), x0 is not explained. What does ‘z is the variable of interest at locations xi and x0‘mean?
Response 4: Usage of disease severity has been changed to disease intensity. X0 and Xi are the sites of observations and Z is the PDI at sites Xo & Xi and accordingly corrected in the manuscript.
Point 5: The organization of the sections 2.5 and 2.6 has to be modified. There is 2.6.1 but no 2.6.2 and part of the information in 2.6 introduction is already in 2.5. Both sections can be merged. Section 2.6.1 could be shortened to avoid repetitions, especially between text and equation.
Response 5: We accept the suggestions and modified accordingly in the manuscript
Point 6: In section 2.8, reorganize to avoid repetitions: the software used (ArcGIS 10.3) is cited 3 times and the loc transformation twice.
Response 6: All suggestions were accepted, accordingly reorganized the section and incorporated suggestions.
Point 7: In section 2.9, line 296: what is district? site, region?
Response 7: District is provinces, site= experiment site (S1, S2….) and Region= Malnad, Coastal, Maidan
Point 8: line 297: if the average PDI is calculated on the 3 regions of study, please write it clearly; if not explain more.
Response 8: Accepted the suggestion and incorporated in the manuscript
Point 9: line 297: if the average PDI is calculated on the 3 regions of study, please write it clearly; if not explain more.
Response 9: Accepted the suggestion and incorporated in the manuscript
Point 10: Lines 307-312: this information is not understandable by non-natives of India. Please clarify.
Response 10: Thirthahalli, Sringeri and Bramhavara are the locations of AHRS belongs to UAHS Shivamogga. This could be difficult to understand by non-natives of India hence deleted the names from the manuscript.
Point 11: Please add the model of regression analysis and the R package used. Explain the confidence intervals that are given as outputs of the analysis. This is not obvious.
Response 11: The information on regression model, R package and outputs has been added as per the suggestion.
Results
Point 1: To first present the PDI across sites, I would propose to only show the maximal PDI of each site each year. This would allow to avoid a mean a long time that is not very meaningful.
Response 1: Your suggestion would be a best idea but it would be better if average PDI represented across the sampling sites in each region. This might provide an information on extent of FRD across the regions.
Point 2: For Maidan area, the map in Fig 6 pointed a zone with lower infection than in the rest of the area. Is it where site 1 is located? In the text, it is not mentioned that S1 has lower infection than other sites (only S2 is said higher than others), but it seems so in Fig 3. In addition, it would be helpful to locate the 9 sites if study in each of the 3 regions.
Response 2: Suggestion has been incorporated in the Manuscript. S1 site had minimum PDI in Maidan region which was revealed in Fig 6 as well as Fig 3.
Point 3: Results presented in the 2 plots (A and B) of Fig 3 are partially redundant and Fig 3 B is redundant with Fig 4 B. They are very difficult to read because not same colors nor scale were used across regions.
Response 3: The redundant figures (3B &B) has been removed to make the manuscript clear and simple.
Point 4: I recommend less figures but bigger so the information can be properly seen: for example, the dynamics in Fig 4B is not visible: larger plots are needed.
Response 4: As per the suggestion, Fig 4B has been removed and remaining figures represented in bigger mode and clearly visible to understand the data.
Point 5: The statement that the 2 years are different is not convincing. Indeed, in Fig 4A, the different statistics plotted
Response 5: Fig 4A clearly represented the variation of FRD across the regions and weeks in both years of experimental study. Revised Fig 4 has been added for better understanding and disease deviation.
Point 6: Line 401: replace ‘Time’ by ‘Year’ because ‘Time’ could also refer to weeks.
Response 6: Corrected in the manuscript
Point 7: Line 416: ’77.14’ should be ’77.84’ according to the Table 4. It is not necessary to give the value for week 10th: 1st and last are enough.
Response 7: Corrected in the manuscript as per the suggestion
Point 8: Section 3.4: the utility of this section is not obvious. Figure 5 is redundant to Figure 4 B from which same results can be inferred (if 2 plots are drawn, one for each year). It does not provide more information about the disease and the regions. I would remove this section to simplify the paper which already contains many different analyses.
Response 8: As per the suggestion, Section 3.4 has been removed from the manuscript to simply the paper
Point 9: Section 3.5.1: the text for Coastal results is not in adequation with the figure 6 B. There might be a problem with the legend and colors. For example, in 2019, according to the figure, 50% is over 51 PDI and 50% over 41 not over 31 and 21 as written in lines 468.
Response 9: According to reviewer’s notice, values and % kriged area has been corrected in the manuscript.
Point 10: For Maidan region, there is more area in orange (21-30) in 2018 than in 2019, contrary to what written on lines 472-473. What those lines mean is not clear. The end of the paragraph, lines 481-483, is not convincing. It seems that general trend across the region is maintained the 2 years.
Response 10: According to reviewer’s notice, values and PDI in Maidan region has been corrected in the manuscript.
Point 11: Section 3.6: the text is difficult to follow because it seems not always related to the table 5 mentioned which show results for the 2 years (not regions).
Response 11: Results of section 3.6 were modified as revealed in ordinary kriging results without regions.
Point 12: Finally, the adding value of calculating the Moran’s I test is not clear from this section. It does not bring any more information than the interpolation approach; or maybe only statistical results. So I would recommend to suppress it or to include the results to support the results of the interpolation.
Response 12: Accepted the suggestion and certainly results of spatial autocorrelation has been modified in support the results of the interpolation.
Point 13: Lines 510-515: what ‘differences’ mean here? I guess it is ‘relationships’ or ‘correlations’ to be used instead. According to Figure 7, the correlations were not always significant, contrarily to what written. The 2 first sentences are very redundant with the 3rd one: this could be shortened and modulated because some correlations are no significant: generalization is not possible.
Response 13: Accepted the suggestion and certainly results of correlation analysis has been modified in the manuscript.
Point 14: The negative links seen between RH and PDI is really surprising. Usually, fungal disease develops with high humidity, which is in adequation with the fact the disease develops after monsoon rains.
Response 14: While analyzing the correlation between PDI and climatic factors, we considered the individual region-wise data set, hence RH was negatively correlated with PDI at Maidan region as the RH level observed in this comparatively lesser than other regions.
Point 15: I would strongly suggest to do these analyses merging the 3 regions. This would allow to enlarge the range of the variables and increase the number of points and may help to get stronger correlations. It would also be interesting to plot the maximal PDI of each of the 27 sites against integrated climatic variables over the 18 weeks period: this would allow deciphering the climatic indicators that could explain the clustering of some areas with lower disease.
Response 15: Repeated the correlation analysis by merging the three regions dataset and also plotted a graph to decipher the impact of climatic indicators across sampling sites.
Point 16: Why not include a region effect in the regression model and try to extract from the whole dataset the climatic variables that influence most the disease? It would be more powerful and avoid to get contrary results from the different regions.
Response 16: Repeated the regression model analysis by adding region effect and thanks for this kind notice, because of that we could get a most favorable results which describes the most influencing climatic indicator for disease occurrence.
Point 17: The interpretation of the table is difficult to follow: the model was not presented in M&M and the outputs are not explained. Which variable gives the significance of the effect? if it is ‘Sig.’ then why is it negative sometimes? and very few effects prove to be significant…
Response 17: As per the suggestions, all the corrections has been modified and corrected in the manuscript.
Discussion
Point 1: The link between disease occurrence and topography is interesting and it’s a pity it was not put in light before. Maybe one easy way would to provide a topographical map of the 3 regions with the 27 sites located on it. This would at least give an idea of the location of the sites and give the opportunity to see the link proposed in lines 582-584.
Response 1: As per the suggestions, a map describing topography three regions and 27 sampling sites has been provided in the manuscript.
Point 2: In accordance with the discussion lines 643-645, it appears necessary to better illustrate the climate diversity at the different sites and the 3 regions. Highlight should be made at the beginning of the paper, on the fact there is lower rainfall at Maidan region.
Response 2: The details of topography, climatic diversity and ecological profile has been provided in Table 1 for better understanding and differentiating the regions studied. It was mentioned in the table that Maidan regions receives annual rainfall of 906.3 mm which comparatively lower than other regions studied.
Point 3: Considering the importance of dew in the morning, it would be interesting to calculate leaf wetness. The same way, taking into account an indicator of inoculum pressure of the sites in the regression model could allow to get more accurate results in a general model including all sites.
Response 3: Thanks for the valuable suggestion, but in case of arecanut its very difficult to record leaf wetness. Due to tall nature (25-30ft height), we couldn’t record the leaf wetness and in monsoon season it’s very difficult to climb the tree due to continuous rainfall as well as quite risky task to take up.
Figures
Point 1: Figure 3: A and B are partly redundant and difficult to read; In Fig 3B X-axis is problematic: sites are called with letters. Replace severity by intensity
Response 1: Redundant figures has been removed and corrected in the manuscript
Point 2: Figure 4: replace severity by intensity. Use same scales of axis when multi-plots to ease the comparisons. Table 4: remove the lines for Difference 1 and 2 that are not useful.
Response 2: According to the suggestions, corrections has been incorporated in the manuscript
Point 3: Figure 7 and Table 6: add the definition of the letters for climatic variables following the titles Table 6: why the model is said ‘spatio-temporal’?
Response 3: Definition of letters used for climatic variables has been added and model name has been changed as Multiple linear regression model
Point 4: Add the signification of the abbreviations of the climatic variables. use same coding as in the text (M&M, Results and Figure 7): RH1 or RH-I…Explain the columns of the table: Sig. for example. If it is ‘significance’, why could it be negative?
Response 4: Corrections has been incorporated in the manuscript
Point 5: English syntax has to be thoroughly revised along the text. Many sentences are not correct.
Response 5: English grammar and syntax has been thoroughly checked along the manuscript
Reviewer 2 Report
The research in the spatio-temporal dynamics and the effect of climate on fruit rot occurrence in areca nut plantations is a good research project, based on a methodology and presenting results, discussion, and relevant and current bibliography. Good manuscript.
Author Response
Author’s Response to Reviewer 2
Dear reviewer
Thank you very much for your kind support and positive response to our manuscript. We appreciate the time and effort that you have rendered to review our manuscript on “Exploring the Impact of Climatic Variables on Arecanut Fruit Rot Epidemic by Understanding the Disease Dynamics in Relation to Space and Time”.